# Molecular Mechanisms of Atrial Fibrillation Recurrence After Successful Catheter Ablation

**DOI:** 10.3390/cells15010036

**Published:** 2025-12-24

**Authors:** Muhammad Sanusi, Roopeessh Vempati, Dinakaran Umashankar, Suha Tarannum, Yash Varma, Fawaz Mohammed, Maneeth Mylavarapu, Faiza Zakaria, Rajiv Nair, Yeruva Madhu Reddy, Christian Toquica Gahona

**Affiliations:** 1Department of Internal Medicine, Trinity Health Oakland/Wayne State University, Pontiac, MI 48341, USA; muhammad.sanusi@trinity-health.org (M.S.); roopeessh.vempati@gmail.com (R.V.);; 2Department of Family Medicine, Western Michigan Homer Stryker M.D. School of Medicine, Kalamazoo, MI 49008, USA; 3Department of Cardiology, Trinity Health Oakland/Wayne State University, Pontiac, MI 48341, USA; 4Department of Cardiology, Endeavor Health Cardiovascular Institute, Glenview, IL 60026, USA; dr.maneeth.mylavarapu@gmail.com; 5Electrophysiology Division, Department of Cardiology, Kansas University Medical Center, Kansas City, KS 66160, USA

**Keywords:** atrial fibrillation recurrence, catheter ablation, left atrial remodeling, fibrosis, connexins and gap junctions, ion-channel remodeling, inflammation and oxidative stress, autonomic nervous system remodeling, genetic susceptibility, MicroRNAs

## Abstract

Atrial fibrillation (AF) is the most common sustained cardiac arrhythmia globally, linked to significant cardiovascular morbidity and mortality. Catheter ablation has emerged as a primary therapeutic approach, yet substantial recurrence rates limit its long-term efficacy. This review critically examines the molecular mechanisms underlying AF recurrence post-ablation, synthesizing recent findings from current literature. Key molecular pathways identified include structural remodeling mediated by fibrosis involving transforming growth factor-beta 1 (TGF-β1) and matrix metalloproteinases (MMPs), ion-channel dysregulation, inflammatory pathways, autonomic nervous system imbalance, and genetic and epigenetic alterations. Despite considerable advances, critical gaps persist due to small, heterogeneous studies and insufficient long-term follow-up. Comprehensive mechanistic research integrating genomics, proteomics, and advanced imaging is urgently needed to better characterize these pathways. Future studies must validate biomarkers such as TGF-β1, MMPs, connexins, and novel markers like GDF-15 and relaxin. Clinical translation of these molecular insights through precision diagnostics and personalized interventions holds great promise to enhance patient selection, optimize ablation strategies, reduce recurrence, and ultimately improve clinical outcomes in AF management.

## 1. Introduction

Atrial fibrillation (AF) is the most prevalent sustained cardiac arrhythmia, impacting more than 33 million people globally [1]. Its occurrence rises significantly with age, and as population demographics shift, the prevalence of AF is projected to more than double within the next 40 years [2]. AF not only doubles the risk of premature death but is also linked to serious cardiovascular complications, including heart failure, stroke, and myocardial infarction [3,4,5,6].

Catheter ablation is a tailored intervention that disrupts the underlying mechanisms driving AF initiation and persistence by eliminating triggers, altering the electroanatomical substrate, and modulating the autonomic nervous system [7]. In the past two decades, catheter ablation for atrial fibrillation (AF) has evolved from a last-resort treatment for patients with severe, drug-resistant symptoms to a widely accepted first-line therapy for individuals with varying types and severities of AF [8,9,10]. Numerous studies have established that catheter ablation is more effective than antiarrhythmic medications for restoring and maintaining normal sinus rhythm [11,12]. Reflecting this evidence, the latest guidelines from the European Society of Cardiology and the American Heart Association/American College of Cardiology/Heart Rhythm Society advocate for catheter ablation in patients with symptomatic atrial fibrillation (AF) [11,12].

While catheter ablation has emerged as a highly effective treatment for atrial fibrillation (AF), its long-term effectiveness is often hindered by considerable recurrence rates [13,14]. This review aims to shed light on the molecular mechanisms underlying AF recurrence after initially successful catheter ablation (Figure 1).

## 2. Structural Remodeling

### 2.1. Fibrosis and Extracellular Matrix (ECM) Alterations

Atrial fibrosis is associated with both the electrophysiological and structural remodeling of atrial tissue and plays a significant role in the development of atrial fibrillation (AF) [15]. While electrical remodeling can be fully restored several weeks after recovery to sinus rhythm, structural remodeling rarely returns to normal, making it more likely for AF to recur. These findings suggest that atrial structural remodeling may play a more critical role in the recurrence and persistence of AF compared to electrical remodeling. Transforming growth factor-β1 (TGF-β1) is a key cytokine that initiates and promotes the synthesis and metabolism of interstitial collagen in the atria. During fibrosis, TGF-β1 can induce the overexpression of extracellular matrix (ECM) proteins, leading to significant changes in cardiac structure and tissue [16,17].

Atrial remodeling is a key characteristic of atrial fibrillation (AF) that creates an irregular environment for electrical conduction, leading to ectopic or re-entrant activity. Transforming growth factor-beta 1 (TGF-β1) plays a crucial role in the atrial remodeling process, including tissue repair and the formation of fibrosis, and is considered a significant factor in causing atrial fibrosis. Studies of atrial specimens from patients undergoing cardiac surgery have shown that TGF-β1 is selectively involved in the development of atrial fibrosis. This occurs through its receptors: TGF-β1 receptor 1 and receptor 2 along with the classical SMAD signalling pathway [18]. Furthermore, the differences observed between atrial and ventricular fibrosis may be attributed to variations in the binding ability of TGF-β1 to its receptors or differences in the phosphorylation levels. Overall, these findings highlight that atrial fibrosis is an important contributor to the onset and maintenance of AF [19].

### 2.2. Role of Transforming Growth Factor-Beta (TGF-β) and Matrix Metalloproteinases (MMPs)

TGF-β1 has been postulated to be a cause of AF by inducing and maintaining AF through the initiation of atrial fibrosis [20]. Collagen, TGF-β1 mRNA, and plasma TGF-β1 concentrations gradually increase in subjects’ sinus rhythm, for both paroxysmal AF and persistent AF, suggesting that plasma TGF-β1 concentrations positively correlate with the degree of atrial fibrosis [21]. Studies in patients with nonvalvular AF have shown that high plasma TGF-β1 concentrations are indicative of low left atrial (LA) endocardial voltage [22]. For example, a study measuring the degree of fibrosis in the LA tissue of patients with persistent AF who underwent open heart surgery for valvular heart disease found that plasma TGF-β1 concentrations correlated with the degree of fibrosis [23]. Moreover, preoperative plasma TGF-β1 concentrations were found to predict the persistence of AF 1 year after surgical maze procedures. Taken together, these findings suggest that TGF-β1 is involved in the mechanisms of atrial fibrosis and AF pathogenesis and that plasma TGF-β1 concentrations may be a surrogate marker of atrial fibrosis. However, plasma TGF-β1 concentrations are also affected by noncardiogenic factors such as the kidneys, lungs, and liver. We therefore measured serum TGF-β1 concentration as an indicator of the level of atrial fibrosis in patients with AF. We found that serum TGF-β1 concentration was significantly lower in patients with AF without recurrence than in those with recurrence, suggesting that serum TGF-β1 concentration correlates with AF recurrence [19].

A group of enzymes capable of degrading almost all ECM proteins, matrix metalloproteinases (MMPs), contribute to both normal and pathological tissue remodeling [24,25,26,27,28,29]. The balance between activated MMPs and tissue inhibitors of metalloproteinases (TIMPs) controls the extent of myocardial ECM remodeling [30,31], including the atrial tissue. However, recent studies have demonstrated the existence of alternative MMP inhibitors [32,33,34,35], namely reversion-inducing cysteine-rich protein with Kazal motifs (RECK), which is thought to be an important regulator of ECM remodeling [32,36,37].

In several cohorts, higher pre-ablation TGF-β1 or MMP-9 levels predicted AF recurrence even after adjustment for left atrial diameter, AF type, and standard clinical risk factors, suggesting incremental prognostic value beyond chamber size alone [19,38,39]. However, most studies did not fully adjust for advanced imaging markers of fibrosis (e.g., LGE-MRI late enhancement or low-voltage area burden), so a substantial part of the association is likely mediated by the degree of underlying atrial myopathy rather than being completely independent. Conceptually, the TGF-β1/MMP axis promotes extracellular matrix accumulation, collagen cross-linking, and myocyte–fibroblast uncoupling. These changes create low-voltage areas and regions of slow, heterogeneous conduction that facilitate re-entry. In this framework, elevated TGF-β1 and MMP-9 levels probably identify more diffuse atrial disease in which both pulmonary-vein reconnection and extra-PV substrate-related circuits are more likely to occur, rather than pointing to a single dominant mechanism of recurrence. It is therefore more accurate to view these biomarkers as integrators of global atrial injury and fibrosis, which interact with procedural factors (durability of PVI, extent of substrate modification) to determine long-term outcome.

Progression of atrial structural remodeling results in the need for more invasive ablation strategies, i.e., complex fractionated electrogram (CFAE) ablation or intra-atrial linear ablation. Despite such aggressive treatment, the post-ablation recurrence rate is higher in these cases than in cases with a less remodeled atrium treated with PVI alone [40,41].

In a study by Okumura et al., Baseline serum MMP-2, carboxylterminal telopeptide of collagen type I (ICTP), and Brain natriuretic peptide (BNP) levels were significantly higher in patients who experienced AF recurrence than in those who did not (860.3 ± 120.8 ng/mL vs. 687.0 ± 122.5 ng/mL; 3.2 ± 1.1 ng/mL vs. 2.7 ± 0.6 ng/mL; 105.0 [38.5–133.0] pg/mL vs. 38.9 [20.8–72.9] pg/mL, *p* < 0.05 for all) [40]. Myocardial collagen turnover biomarker levels are associated with post-ablation AF recurrence. A 2-month post-ablation elevation in MMP-2 levels suggests that wound healing persists for that long after ablation. The levels of systemic and intracardiac biomarkers of both inflammation and collagen turnover were similar, which implies that these serum biomarkers may originate in the systemic circulation rather than in the heart [42].

### 2.3. Cellular Hypertrophy and Myocyte Disarray

Myocyte hypertrophy and cellular disarray, as seen in patients with hypertrophic cardiomyopathy (HCM), can also increase the risk of recurrence of atrial fibrillation. Atrial myocyte hypertrophy increases LA wall thickness, thus reducing the likelihood of creating durable transmural lesions. HCM is commonly associated with LA dilatation, remodeling, and diffuse interstitial fibrosis, thus increasing the risk of elicitation of triggering from extra-pulmonary vein foci [43]. Additionally, a multicenter study found that most patients require re-do procedures, that efficacy was obtained with the concomitant administration of anti-arrhythmic drugs (AADs) in most patients, and that efficacy was time-dependent [44]. The probability of post-ablation recurrence of atrial arrhythmias was found to be two-fold higher in patients with HCM compared with that in controls, after either single or multiple procedures. AADs were found to be paramount in maintaining sinus rhythm, as shown by a five-fold higher probability of AAD use after ablation in patients with HCM. Nonetheless, the combination of catheter ablation with AADs was associated with a success rate as high as 75% after a single procedure [43,45,46,47]. Treatment with AADs in HCM patients is less effective than in patients without HCM. Appropriate selection of specific AADs can overcome this limitation. Amiodarone and sotalol are the most frequently used AADs in this clinical setting, the former being the most effective and the latter being better tolerated. Dofetilide is also well tolerated, and recent data suggest a high efficacy during long-term follow-up [43].

### 2.4. Impact on Conduction Velocity and Arrhythmogenic Substrate Formation

AF substrate is, at least in part, dynamic, with rate-dependent effective refractory periods and conduction velocities. These electrophysiological factors can lead to the so-called functional phenomena. The atrium, under higher rates and conduction velocities, is more prone to functional phenomena than the ventricle, leading to complex electric wavefront dynamics during AF [48]. Frontera, A et al. summarized that dual high-density mapping during sinus rhythm and after an atrial extra stimulus reveals differences in areas with conduction abnormalities, particularly represented by pivot sites and zones of slow conduction. The change in direction and velocity of the traveling wavefront, when an extra stimulus is delivered with a short coupling interval, uncovers “functional” rhythm-dependent conduction abnormalities that can predict atrial fibrillation (AF) recurrences during follow-up. Functional conduction abnormalities are found at sites with preserved voltage and are often associated with anatomical structures such as the esophagus and the sinus of Valsalva in the ascending aorta. Pulmonary vein isolation (PVI) remains the primary treatment for AF; however, this procedure does not address the entire complexity of AF pathogenesis. Success rates for PVI typically range from 50% to 70% in patients with persistent AF (PsAF), and a considerable number of patients experience recurrent AF episodes. As a result, current treatment options for these patients are still limited. This study highlighted a new methodology for characterizing the atrial substrate in patients with AF. The mapping protocol performed during sinus rhythm, combined with the application of an extra stimulus delivered from the coronary sinus, is both easily reproducible and reliable in any electrophysiology laboratory. Identifying slow conduction sites and pivot points during either sinus rhythm or paced rhythm can provide a better characterization of the substrate that sustains localized reentrant circuits [49].

While catheter ablation for paroxysmal AF yields satisfying outcomes in terms of AF recurrence, persistent AF remains associated with high recurrence rates. Therefore, patients with persistent AF may require different ablation strategies [50]. Lukito et al., in their meta-analysis, investigated and performed a diagnostic test meta-analysis on whether slow left atrial conduction velocity (LACV) in the anterior wall calculated by electroanatomic mapping predicts atrial fibrillation (AF) recurrence after catheter ablation. Seven studies involved a sample size of 1428 patients, with a mean follow-up duration of 13 months. Patients with AF recurrence have slower LACV in the anterior wall (mean difference − 0.16 m/s [−0.18, −0.15], *p* < 0.001). Slow LACV in the anterior wall, defined as LACV below 0.70–0.88 m/s, was associated with increased AF (adjusted OR 3.41 [1.55, 7.50], *p* = 0.002). Slow LACV in the anterior wall has an AUROC of 0.80 [0.76–0.83], sensitivity of 70% [52, 84], specificity of 76% [67, 83], positive likelihood ratio of 2.9 [2.3, 3.6], negative likelihood ratio of 0.39 [0.25, 0.63] for predicting AF recurrence post ablation [50]. Tissue fibrosis occurs due to various insults, leading to intercellular gap junction remodeling and abnormal distribution, which results in altered conduction and atrial strain [51,52]. Myocardial fibrosis promotes remodeling, which increases the resting membrane potential of myocytes, causing decreased conduction velocity and resulting in shortened action potential duration and refractory period. These conduction disturbances may contribute to the development of re-entrant circuits in AF. Hence, slow LACV may serve as an indicator of left atrial remodeling or fibrosis and is closely related to the development and recurrence of AF [53]. Low voltage areas and slow conduction velocities more accurately depict atrial fibrosis and interatrial conduction block, which are closely associated with conduction issues and an increased risk of AF recurrence [54]. Determining the association between slow LACV and AF recurrence may pave the way for utilizing high-density mapping systems and noninvasive cardiac studies to detect slow conduction by identifying prolonged P-wave duration or regions of potential re-entry, such as left atrial heterogeneity through late gadolinium enhancement (LGE). This may improve the prognostication of AF recurrence post-PVI [55]. High-density mapping studies show that zones of slow conduction and pivot points tend to co-localize with areas of LGE-MRI fibrosis and low-voltage substrate in the left atrium [56,57]. A greater burden in these regions has been associated with higher recurrence rates and with inducible organized atrial tachycardias after ablation, particularly in patients with persistent AF. These observations have been reproduced across different mapping platforms and multipolar catheters, although the absolute thresholds used to define “low voltage” and “slow conduction” vary with catheter geometry, filtering, and pacing protocol. At present, left atrial conduction velocity, pivot zones, and slow-conducting areas are best considered qualitative markers of underlying atrial myopathy that can complement imaging and voltage mapping, rather than fully standardized metrics that can be applied interchangeably across all systems.

### 2.5. Ion-Channel Dysregulation

The mechanisms underlying AF induction and maintenance are incompletely understood, but it is generally accepted that re-entry is the major mechanism of AF maintenance. Re-entry induction requires an appropriate vulnerable substrate, as well as a trigger that initiates re-entry within the substrate. Long-term AF causes profound alterations in atrial structure (cardiomyocyte hypertrophy, glycogen accumulation, and interstitial fibrosis). These lead to inhomogeneous conduction slowing that promotes the development of anatomically fixed re-entry circuits. Atrial remodeling is clinically important as it indicates the large resistance of persistent AF to treatment [58].

### 2.6. Electrophysiological and Structural Determinants

Atrial fibrillation (AF) recurrence following catheter ablation (CA) is influenced by multiple electrophysiological and structural factors, including left atrial (LA) asynchrony, abnormal atrial voltage, and altered conduction patterns. Emerging evidence suggests that heterogeneity in atrial conduction and remodeling of the LA substrate contribute significantly to post-ablation arrhythmia recurrence [59].

Recent studies utilizing pulsed-wave tissue Doppler imaging (PW-TDI) have demonstrated that LA asynchrony is a key determinant of AF recurrence [60]. Two novel indices, the difference between the longest and shortest P-A0 intervals (DLS) and the standard deviation of activation times across four LA sites (SD4), have been proposed to quantify asynchrony. Notably, patients experiencing AF recurrence exhibited significantly higher DLS (43 ± 16 ms vs. 35 ± 16 ms; *p* = 0.01) and SD4 (20 ± 8 ms vs. 16 ± 7 ms; *p* = 0.004) than those without recurrence. Additionally, LA scarring, as indicated by low-voltage areas (<0.5 mV) on electroanatomical mapping, was associated with prolonged DLS (46 ± 19 ms vs. 36 ± 16 ms; *p* = 0.04) and SD4 (23 ± 9 ms vs. 16 ± 7 ms; *p* = 0.009). These findings underscore the role of conduction heterogeneity and delayed atrial activation in sustaining AF and increasing the likelihood of post-ablation recurrence. Structural remodeling of the LA further contributes to AF persistence and recurrence. Among patients undergoing electroanatomical mapping, all individuals with low-voltage areas exhibited a distinct “U-pattern” of atrial activation, characterized by conduction propagating from the inferior to anterosuperior regions. In contrast, patients with normal LA voltage demonstrated a D-pattern of activation, consistent with conduction through Bachmann’s bundle. The strong association between the U-pattern, LA scarring, and AF recurrence suggests that interatrial conduction block and altered atrial excitation patterns play a pivotal role in post-ablation outcomes [60].

Multivariate analysis further supported these findings, identifying DLS (HR: 1.02, *p* = 0.008) and SD4 (HR: 1.35, *p* = 0.031) as independent predictors of AF recurrence and the presence of atrial scarring [61]. Interestingly, LA size (*p* = 0.08) and age (*p* = 0.35) were not significant predictors, highlighting the importance of functional conduction abnormalities over structural enlargement in determining ablation success [61]. These observations align with prior research demonstrating that total atrial conduction time, as assessed by tissue Doppler imaging, is a reliable predictor of AF recurrence following CA.

From a clinical perspective, these findings suggest that pre-procedural assessment of LA asynchrony and conduction abnormalities could enhance risk stratification and guide ablation strategies. The presence of delayed anterior LA activation and conduction block across Bachmann’s bundle may serve as markers of advanced atrial remodeling, warranting a tailored approach to ablation in these patients [60]. Although the study had limitations, including a relatively small cohort of persistent AF cases and individuals with extensive LA scarring, it provides a novel electrophysiological framework for predicting post-ablation outcomes and optimizing patient selection.

### 2.7. Role of Connexins and Gap Junctions

AF recurrence following CA is influenced by electrophysiological and structural remodeling of the atria [62]. One critical component of this process is the gap junctional network, which facilitates impulse propagation between cardiomyocytes. Gap junctions, composed of connexins (Cxs), play a crucial role in maintaining synchronized atrial activation. Disruptions in connexin expression, localization, and function have been linked to increased AF susceptibility, conduction heterogeneity, and reentrant activity, which may contribute to post-ablation recurrence [62]. Gap junctions in the atria primarily comprise Cx40, Cx43, and Cx45, with Cx40 being the predominant connexin in atrial tissue [63]. Remodeling of these connexins has been widely observed in AF, with Cx40 and Cx43 displaying altered expression patterns, lateralization from intercalated disks, and abnormal hemichannel activity [64,65]. Cx40 downregulation or redistribution has been strongly associated with conduction heterogeneity and AF recurrence, supporting its role in sustaining arrhythmic substrates [66]. Multifaceted mechanisms contribute to connexin remodeling and AF recurrence, including gene mutations, transcriptional dysregulation, and post-translational modifications. Several mutations in the GJA5 gene encoding Cx40 have been identified in idiopathic and familial AF cases, leading to impaired gap junction function and reduced electrical coupling [67]. Similarly, Cx43 mutations (GJA1) have been linked to conduction slowing and reentrant arrhythmias, suggesting that dysfunctional connexin interactions may create a substrate for AF persistence [68]. Beyond genetic alterations, connexin phosphorylation and trafficking defects contribute to AF-associated gap junction remodeling. Increased lateralization of Cx40 and Cx43 from intercalated disks has been documented in chronic AF, resulting in abnormal conduction and increased reentry [65,69]. These findings indicate that gap junction remodeling, rather than total connexin depletion, plays a predominant role in AF recurrence. The implications of connexin dysfunction extend to therapeutic strategies aimed at reducing AF recurrence post-ablation. Beta-blockers such as metoprolol have been shown to mitigate gap junction remodeling, restoring normal Cx43 distribution and reducing conduction heterogeneity [69]. Moreover, connexin gene therapy and pharmacological modulation of hemichannel function have been explored as potential AF interventions [70].

In summary, connexin dysregulation and gap junction remodeling play a pivotal role in AF recurrence after catheter ablation. The loss of coordinated atrial conduction due to altered Cx40 and Cx43 expression, mislocalization, and impaired electrical coupling provides an arrhythmogenic substrate that sustains AF. Targeting gap junction integrity and connexin function represents a promising approach to improving long-term outcomes in post-ablation AF patients.

## 3. Inflammatory Pathways

Inflammation plays a key role in recurrent atrial fibrillation (AF) following ablation. Mechanical and surgical injury triggers the release of inflammatory mediators, including cytokines and chemokines, leading to atrial remodeling and sustaining AF [71]. Rosenberg et al. highlight that ablation therapy increases damage-associated molecular patterns (DAMPs), inflammatory cytokines, immune cell activity, and collagen remodeling, contributing to AF recurrence. DAMPs, such as HMGB-1, mtDNA, heat shock proteins (HSPs) 27, 70, and 60, along with the NLRP3 inflammasome, activate toll-like receptors (TLRs), contributing to atrial remodeling via initiation of sterile immune responses [72].

Several studies have examined individual inflammatory mediators in AF recurrence. Cao et al. identified the TGF-β1 C-509T polymorphism as an independent predictor of AF recurrence, with elevated TGF-β1 promoting atrial fibrosis [73]. Tian et al. also linked higher serum TGF-β1 to extensive fibrosis and increased AF recurrence risk [19]. Wei et al.’s study of 150 AF patients undergoing radiofrequency catheter ablation found that elevated growth differentiation factor-15 (GDF-15) levels were associated with left atrial remodeling and higher recurrence rates [74]. Zakynthinos et al. highlighted the role of endothelial dysfunction in AF persistence, noting that higher asymmetric dimethylarginine (ADMA) levels—an inhibitor of nitric oxide synthase (NOS)—are linked to increased AF recurrence [75].

Inflammatory cytokines such as CRP, TNF-α, IL-2, and IL-6 contribute to recurrent AF by recruiting PMNs, macrophages, and fibroblasts, driving ECM remodeling. Angiotensin-II-stimulated TGF-β1 further promotes collagen deposition, leading to atrial fibrosis. Systemic inflammation thus plays a critical role in structural remodeling, fibrosis, and conduction abnormalities, sustaining AF persistence [72]. Endothelial dysfunction contributes to fibrosis and electrical remodeling, further promoting atrial substrate modifications that favor AF initiation and maintenance [75]. Most studies that evaluated inflammatory biomarkers in the ablation setting followed a common sampling pattern: blood was drawn at baseline, repeated at 24–72 h after the procedure, and in some cases at 1 week and 3 months. High baseline hs-CRP and IL-6 and larger early post-procedural rises have been associated with a greater risk of AF recurrence, particularly within the first 3–6 months after ablation [76,77,78,79]. CRP levels often peak around days 2–3 after RF ablation and then decline, whereas patients who maintain sinus rhythm tend to show a more pronounced long-term decrease in inflammatory markers. By contrast, GDF-15 and relaxin have been studied mainly as baseline risk markers. Elevated pre-ablation GDF-15 seems to reflect a higher overall cardiovascular risk burden and has shown inconsistent associations with AF incidence and recurrence [80]. Baseline serum relaxin has been more consistently linked to AF recurrence after RF ablation, although data are still limited to small cohorts [81,82]. Systematic data on serial GDF-15 and relaxin measurements over time, and how their trajectories relate to late (>12 months) recurrence, are still lacking. Differences between energy sources may influence biomarker profiles. RF ablation typically causes larger and more prolonged CRP and IL-6 elevations than cryoballoon ablation in some series, whereas robust comparative data for PFA are only now emerging from ongoing trials [83,84,85]. Taken together, these observations suggest that baseline inflammatory status and early post-procedural inflammatory response provide complementary information about short- and intermediate-term recurrence risk, but we still lack a validated, energy-specific biomarker algorithm to guide clinical decisions.

Multiple studies [71,72,75,86] highlight oxidative stress as a key driver of atrial remodeling and recurrent AF. Reactive oxygen species (ROS) induce cellular damage, alter ion channels, and disrupt atrial electrical properties. Major ROS sources in atrial tissue include mitochondrial dysfunction, NADPH oxidases, and uncoupled nitric oxide synthases. Polymorphonuclear cells (PMNs) also release myeloperoxidase (MPO) and activate NADPH oxidase, generating free radicals through nitric oxide (NO) uncoupling. This cascade promotes atrial fibrosis, structural remodeling, and AF progression.

## 4. Autonomic Nervous System (ANS) Imbalance

Imbalances in the autonomic nervous system (ANS), including heightened sympathetic and parasympathetic activity, disrupt atrial electrophysiology and contribute to AF [72,86]. Choi et al. highlight the intrinsic cardiac autonomic system, particularly the ganglionated plexi, in modulating atrial electrophysiology, where imbalances trigger atrial tachyarrhythmia and elevate AF risk [87].

Molecular mediators like nerve growth factor (NGF) and catecholamines play key roles in autonomic nervous system (ANS) remodeling, contributing to AF pathogenesis. NGF overexpression enhances sympathetic innervation, altering atrial electrophysiology and creating a substrate for AF. Excessive catecholamine release, particularly norepinephrine, increases adrenergic signaling, promoting triggered activity, shortening refractory periods, and increasing repolarization dispersion. These changes drive atrial remodeling, increasing AF susceptibility and persistence [71].

## 5. Genetic and Epigenetic Factors

The cardiac action potential, a marvel of biological engineering, forms the basis of the cardiac conduction system and is primarily influenced by the intricate shift of sodium, calcium, and potassium ions. This complex exchange of biological molecules, orchestrated by intercalated disks between myocytes, plays a crucial role in propagating the action potential [88].

Several potassium channels, including the delayed-rectifier potassium current, IKs, are expressed in cardiac cells. These channels are responsible for maintaining the resting membrane potential, a crucial factor in the heart’s electrical activity. They play a vital role in the different repolarization phases. The KCNQ1 gene encodes the cardiac IKs channel, and it was the first disease gene linked to adult-onset familial atrial fibrillation (AF). The KCNA5 gene encodes Kv1.5, a component of the voltage-gated IKur potassium channel. KCNA5 is a promising candidate gene for familial AF due to its atrium-specific functional expression [89]. Other gain-of-function mutations have also been identified in the KCNE1, KCNE2, KCNE, KCNE5, KCNQ1, and KCNJ2 genes as causes of AF. Furthermore, mutations in the SCN5A gene, which encodes a cardiac sodium channel, have also been linked to this condition [90].

The *MYL4* gene is responsible for the atrium’s electrical, contractile, and structural integrity. It plays a crucial role in the normal functioning of the atrium, ensuring its proper electrical activity, contractility, and structural stability. A loss-of-function variant in the MYL4 gene was identified as causing early atrial fibrosis, which in turn leads to atrial cardiomyopathy and atrial arrhythmia. This gene is a key player in maintaining the atrium’s health, and any variations in its function can lead to atrial fibrillation [91]. Titin (*TTN*) is considered a critical AF gene, as previously mentioned. The gene encodes a giant sarcomere protein, titin, expressed in all chambers of the human heart. Titin-truncating variants (TTNtv) have been shown to predispose individuals to AF directly [92].

Catheter ablation is a commonly used technique to manage rhythm in atrial fibrillation (AF). However, for many patients, it may not be beneficial due to the recurrence of AF. Since it is an intervention with several health risks, predicting which patients are more likely to benefit from the procedure would be advantageous. Although still in its early stages, the potential of genetic risk scores, along with other relevant factors, to aid physicians in determining whether to perform catheter ablation on patients is a promising development. Polymorphisms in the chromosome locus 4q25, which contains two single-nucleotide polymorphisms (SNPs) (rs2200733, shown to be most strongly associated with atrial fibrillation (AF), and rs10033464, near pituitary homeobox 2 (PITX2)), have been shown to influence the risk of AF recurrence after catheter ablation [93]. Genome-wide association studies (GWASs) have been conducted to identify these variations, and PITX2 is a transcription factor that plays a crucial role in the embryologic development of the heart, the suppression of sinus node formation in the left atrium [94], and the formation of the pulmonary vein myocardial sleeves [95]. It has been found to impact AF recurrence significantly [96].

Several circulating microRNAs (miRNAs) have been explored as predictive biomarkers of AF recurrence after ablation. Small studies suggest that miR-21, miR-29, and newer candidates such as miR-451a and miR-15a-5p may differentiate patients who relapse from those who remain in sinus rhythm, but findings are heterogeneous, and sample sizes are modest [97]. MiR-1, a muscle-specific microRNA (miRNA) and the most abundantly expressed miRNA in both ventricles and atria, has been shown to exert a proarrhythmic effect by modulating the inwardly rectifying potassium channel subfamily J member 2 (KCNJ2) through enhanced expression [98]. This upregulation is significantly observed in cardiomyocytes from patients with atrial fibrillation (AF) [99]. Importantly, multiple studies have also demonstrated that miRNAs are involved in various cardiovascular processes, including cardiac remodeling, fibrosis, inflammation, and arrhythmogenesis [100,101]. They have proven to be attractive candidates as biomarkers due to their stability in circulation and tissue-specific expression patterns [102,103]. Zhou Q. et al. revealed that plasma miR-21 levels are strongly correlated with left atrial low-voltage areas (LVAs), which are directly indicative of atrial fibrosis. Patients with lower miR-21 levels showed better outcomes after ablation, suggesting that it may serve as a potential predictor and marker of ongoing atrial remodeling [104]. The proarrhythmic effect of MiR-1 emphasizes its potential as a promising target for further research into the role of miR-1 in AF. Beyond miR-1 and other ion-channel-modulating miRNAs, AF-related atrial fibrosis appears to reflect an imbalance between pro-fibrotic and anti-fibrotic microRNAs. Pro-fibrotic miRNAs, such as miR-21, miR-132, miR-199, miR-208, and miR-34a, promote fibroblast activation, collagen synthesis, and TGF-β/Smad signaling, whereas anti-fibrotic miRNAs, such as the miR-29 family, miR-30, miR-133, and miR-101, restrain extracellular matrix deposition and fibrotic gene expression [105].

However, in the context of AF, global DNA methylation levels are significantly increased in AF patients, positively correlating with age and suggesting a role in cardiac inflammation and fibroblast activation [106,107]. Histone deacetylation, catalyzed by distinct classes of histone deacetylases (HDACs), is linked to gene silencing [108]. Emerging evidence suggests a crucial role for HDACs in regulating the post-transcriptional expression of various proteins in cardiomyocytes within the context of atrial fibrillation (AF), particularly those related to cytoskeletal and conductive proteins [109]. Conversely, their role in contractile and ion channels remains unclear. However, the potential of HDAC inhibition to significantly block or halt the progression of AF is a promising development, further underscoring the importance of HDAC as a potential therapeutic target in AF [109]. Nonetheless, the molecular mechanisms require further exploration. Further methylation analysis of left atrial tissue from patients with permanent atrial fibrillation (AF) revealed 417 differentially methylated cytosine–phosphate–guanine (CpG) sites involved in inflammation activation, sodium and potassium ion transport, fibrosis, and the reduction in lipid metabolism [110]. Hypermethylation of DNA suppresses the expression of antiproliferative and anti-myofibroblast differentiation genes in the human heart, accelerating cardiac fibrosis and AF recurrence [111,112].

## 6. Neurohumoral Factors and Circulating Biomarkers

The sympathetic co-transmitter neuropeptide Y (NPY) is a promising biomarker for sympathetic hyperactivity following myocardial infarction [113] and heart failure [114]. Increasing evidence indicates that the autonomic nervous system plays a role in inducing AF. NPY has been recognized as a biomarker of neuromodulation in AF through its receptors in the left atrium [115]. A study comparing persistent or long-standing atrial fibrillation (AF) with paroxysmal AF suggested a link between AF progression and left atrial NPY receptor expression [116]. This indicates that neuromodulation is present. Song et al. concluded that NPY could inhibit fibrosis by reducing apoptosis in atrial myocytes and suppressing the activation of Akt (protein kinase) in cardiac fibroblasts, consequently increasing the incidence of AF post-ablation [117]. Circulating NPY concentrations are higher in patients with AF than in those in sinus rhythm, and levels tend to increase with AF persistence and symptom burden [118]. Elevated pre-operative NPY has also been associated with an increased risk of post-operative AF after cardiac surgery [119]. Experimental work in canines suggests that long-term changes in NPY signaling after circumferential PV ablation may influence AF inducibility and atrial remodeling, but these findings have not yet translated into routine peri-ablation monitoring of NPY in humans [119].

Substance P (SP) is an 11-amino acid peptide that belongs to the tachykinin family of neuropeptides [120]. SP is primarily found in C-fiber sensory nerves that innervate the coronary arterial system, the intrinsic cardiac neural plexus, intrinsic nerve bundles, and interganglionic nerves in the atria, as well as occasionally in cardiomyocytes [120]. Yu et al. paced dog hearts at 600 beats per minute to induce atrial fibrillation and discovered that, while there was an increase in sympathetic nerve fibers in the atria, there was a significant loss of substance P (SP)-containing nerve fibers [120]. Given that most cardiac sympathetic (SP) activity is in the atria and that SP can influence heart rate, leading to bradycardia, it is reasonable to conclude that a loss of SP could contribute to atrial fibrillation. Neuromodulatory strategies that target cardiac ganglionated plexi or enhance vagal tone (e.g., GP ablation, low-level vagus nerve stimulation, ethanol infusion into the vein of Marshall) have shown that acute changes in autonomic tone can impact AF initiation and maintenance [121,122,123]. Small mechanistic studies have reported that biomarkers such as NPY, S100B and other nerve-related proteins can reflect acute autonomic injury during AF ablation and might predict maintenance of sinus rhythm [116,124,125]. However, dynamic measurements of NPY or substance P before and after neuromodulatory procedures remain sparse, and their direct value for predicting long-term AF recurrence after catheter ablation is unproven. Autonomic neuropeptides should therefore be viewed as promising mechanistic markers rather than established clinical tools at this time.

Two biomarkers reflect and modulate AF: N-terminal pro-B-type natriuretic peptide (NT-proBNP) [126] and high-sensitivity C-reactive protein (hs-CRP) [127]. Both biomarkers have been shown to predict AF recurrence, irrespective of radiofrequency catheter ablation of AF (RFCA). A multicenter study by Carballo et al. found that NT-proBNP levels at 6 months post-RFCA were significantly higher in patients with AF recurrence compared to those without (*p*  =  0.001, Wilcoxon rank sum test) [78]. Additionally, a meta-analysis indicated that elevated high-sensitivity C-reactive protein and Interleukin-6 are also strong markers of AF recurrence [76].

The left atrial appendage (LAA) is a remnant of the embryonic left atrium, while the smooth-walled left atrium arises from the primordial pulmonary veins and their branches. The LAA is an actively contracting structure, and studies have shown that it is more distensible than the left atrium [128]. As a result, the LAA may help maintain left atrial pressure by activating stretch-sensitive receptors and influencing the actions of the atrial natriuretic factor (ANF), which increases heart rate, diuresis, and natriuresis [129]. Atrial arrhythmias, such as atrial fibrillation (AF), are known predisposing factors for thrombus formation, with the left atrial appendage (LAA) serving as a primary site. The function and morphology of the LAA are believed to contribute to the origin of thrombus. Echocardiographic measurements of LAA function are directly correlated with the risk of thromboembolism. Lower LAA velocities have been associated with ischemic stroke and the formation of thrombi. Three known morphologic features of the LAA linked to ischemic stroke are its shape, orifice size, and fibrosis [130].

Growth differentiation factor-15 (GDF-15) is a member of the transforming growth factor β (TGF-β) cytokine family, which is widely distributed in mammalian tissues, including the prostate, intestinal mucosa, and kidney. It has been shown to play multiple roles in various pathologies, such as cardiovascular disease [131]. A study by Kempf et al. found that GDF-15 contributes to cardiomyocyte repair in response to tissue inflammation, oxidative stress, and injury, exhibiting both anti-apoptotic and anti-hypertrophic properties [132]. Suppression of Tumorigenicity-2 (ST-2) belongs to the interleukin-1 (IL-1) receptor family and can be used in conjunction with GDF-15 as a marker of cardiac stress. It has two forms, soluble ST2 (sST-2) and transmembrane ST-2 (ST-2L), both of which are upregulated in response to myocyte stretch, similar to brain natriuretic peptide (BNP) [133]. Wei et al. found that the expression of GDF-15 in serum was elevated in the persistent AF group compared to the paroxysmal AF group and also increased in the recurrence group compared to the non-recurrence group; however, it decreased after RFCA [74]. This suggests the potential of using GDF-15 to predict AF recurrence after ablation. In a study by Gizatulina et al., sST2 was identified as the only independent predictor of AF recurrence. A 10-unit increase in sST2 was associated with a 2.10-fold increase in the risk of AF recurrence. Serum sST2 levels at a cut-off value of 36 ng/mL or greater can serve as a predictor [134].

Relaxin, a peptide hormone of the insulin superfamily, promotes the remodeling of the extracellular matrix [135]. It has emerged as a natural suppressor of age-related fibrosis in various tissues, including the skin, lungs, kidneys, and heart. Several downstream pathological processes are involved, including reduced expression of transforming growth factor-β (TGF-β) and tumor necrosis factor-α (TNF-α), as well as increased activity of matrix metalloproteinases (MMPs), which leads to decreased fibrosis [135]. Hao Zhou et al. demonstrated that relaxin is associated with fibrosis-related biomarkers and is significantly elevated in atrial fibrillation (AF) [136]. A study by Qu et al. demonstrated that elevated pre-RFCA relaxin levels are associated with post-RFCA AF recurrence [82]. The exact mechanism by which relaxin contributes to atrial fibrillation (AF) recurrence is not fully understood, but it may be related to its effects on atrial remodeling and fibrosis.

## 7. Emerging Exposures

Emerging research links microplastics and nanoplastics (MNPs) to systemic inflammation, oxidative stress, and endothelial dysfunction, key contributors to cardiovascular diseases like AF [137]. MNPs trigger inflammation, activating immune cells and causing genomic instability, which leads to vascular endothelial dysfunction [138]. A landmark *New England Journal of Medicine* study reported that patients with MNPs in carotid plaque had roughly a 4–5-fold higher risk of myocardial infarction, stroke, or death compared with those without detectable plastics, supporting a link between plastic contamination of the vasculature and adverse cardiovascular events [139]. However, specific studies linking microplastics to AF recurrence post-ablation are currently lacking. Given the established role of inflammation and oxidative stress in AF pathogenesis, further research is warranted to explore potential connections between microplastic exposure and AF outcomes.

## 8. Ablation Energy Sources and Their Biological Footprint

Radiofrequency and cryoballoon ablation create lesions by heating or freezing tissue, whereas pulsed field ablation (PFA) uses short, high-voltage electrical pulses to cause non-thermal electroporation of cell membranes. Preclinical and early clinical data show that PFA is relatively selective for cardiomyocytes, while largely sparing nearby structures such as the esophagus, pulmonary veins, and phrenic nerve, although rare phrenic nerve injury has been reported [85,140,141]. Randomized trials and meta-analyses demonstrate that PFA is non-inferior to conventional thermal ablation for 1-year freedom from atrial arrhythmias, with similar acute pulmonary vein isolation rates and shorter procedure times [142,143]. Repeat-procedure data suggest that lesion durability and pulmonary vein reconnection rates are comparable between PFA and high-power radiofrequency ablation, with some series showing numerically lower reconnection after PFA, although differences are modest and still under investigation [144,145,146]. In terms of systemic response, RF and cryoablation trigger marked rises in inflammatory and myocardial injury markers (hs-CRP, IL-6, fibrinogen, and troponin), which relate to early recurrence [79,147]. Early work suggests that PFA may produce a different inflammatory and biomarker profile, potentially reflecting its non-thermal mechanism, but head-to-head data are still limited, and ongoing studies are specifically addressing this question [84].

## 9. Therapeutic Implications

### 9.1. Colchicine

Colchicine was introduced to cardiology nearly four decades ago for treating recurrent pericarditis [148]. In recent years, its role has expanded to other cardiovascular conditions. Atrial fibrillation (AF) is the most common complication following cardiac surgery. Given colchicine’s efficacy in preventing pericardiotomy syndrome, several trials have evaluated its role in preventing postoperative atrial fibrillation (POAF). A systematic review and meta-analysis demonstrated a significant reduction in POAF (RR: 0.70, 95% CI: 0.58–0.84, *p* = 0.0001) [149]. However, only two of the six included trials showed a positive effect, while four showed neutral results. Data on hospital length of stay were inconsistent [150,151], and no study has demonstrated a reduction in all-cause mortality.

Ablation procedures for AF trigger a proinflammatory response, contributing to early AF recurrence [152]. Colchicine has shown benefit in reducing AF recurrence at 3- and 12-month post-ablation (RR: 0.57, 95% CI: 0.39–0.83, *p* = 0.0032; RR: 0.58, 95% CI: 0.42–0.80, *p* = 0.0008) [149,153]. Larger trials are needed to confirm colchicine’s effectiveness in secondary AF prevention.

### 9.2. Corticosteroids

Corticosteroids, which modulate multiple inflammatory pathways, have been studied in both POAF and AF recurrence after catheter ablation (CA). Koyama et al. reported a reduction in immediate AF recurrence (≤3 days post-PVI) with low-dose corticosteroids for three days; however, recurrence between 4 and 30 days was unchanged. At 14 months, AF-free survival was higher in the corticosteroid group (85% vs. 71%, *p* = 0.032) [154]. In contrast, a study by Won et al. found no significant reduction in recurrence with pre-procedure IV corticosteroids [155].

IV corticosteroids have been shown to alter levels of IL-6, IL-8, IL-10, CRP, nitric oxide, endothelin-1, and complement C4 [156]. One study found that IV hydrocortisone before PVI increased the need for radiofrequency ablation and dormant PV conduction [157]. Despite some positive signals, the role of corticosteroids in secondary AF prevention remains uncertain due to their potential adverse effects, such as GI bleeding and infections.

### 9.3. Methotrexate

Methotrexate is associated with a lower incidence of new-onset AF in patients with seropositive rheumatoid arthritis [158]. Though the exact mechanism is unclear, its anti-inflammatory properties may contribute to risk reduction.

### 9.4. Cardiac Drugs with Pleiotropic Effects

Renin–angiotensin system (RAS) inhibitors have shown benefits in both primary and secondary AF prevention. Meta-analyses indicate reduced recurrence after electrical cardioversion and in patients with paroxysmal AF [159].

Baia-Bezerra et al. demonstrated that sacubitril–valsartan (SV) significantly reduced AF recurrence post-CA compared to ACEi/ARB (RR: 0.54; 95% CI: 0.41–0.70; *p* = 0.000004; I^2^ = 80%) [160]. Similarly, Sun et al. found that RAS inhibitors lowered AF recurrence post-ablation (RR: 0.85; 95% CI: 0.72–0.99; *p* = 0.03), with SV showing even greater benefit (RR: 0.50; 95% CI: 0.37–0.68; *p* < 0.00001) [161].

Although a specific mechanism is not well defined, losartan has been shown to reduce stress-induced electrical remodeling in atrial cardiomyocytes [162], and Goette et al. reported increased Erk1/Erk2 and ACE expression in AF patients [163], suggesting a role in atrial fibrosis.

### 9.5. Statins and PCSK9 Inhibitors

Statins have been shown to reduce AF recurrence after CA, independent of LDL-lowering effects [164]. Mechanistically, they reduce lysyl oxidase, collagen deposition, and cross-linking—key processes in atrial remodeling [164].

While PCSK9 inhibitors have not been studied in randomized AF trials, preclinical data suggest that evolocumab may reduce AF susceptibility in animal models [165]. However, a clinical trial found that alirocumab did not significantly affect AF incidence post-acute coronary syndrome (HR: 0.91; 95% CI: 0.77–1.09) [166].

### 9.6. Metformin

Metformin activates AMPK and inhibits fibroblast differentiation and atrial fibrosis, key substrates for AF [167]. An observational study showed it reduced AF and ventricular arrhythmias compared to sulfonylureas [168]. The ongoing phase 4 trial (NCT04625946) is evaluating metformin’s role in preventing AF recurrence post-CA [169].

### 9.7. Sodium–Glucose Cotransporter 2 Inhibitors

Sodium–glucose cotransporter 2 inhibitors (SGLT2i) reduce AF recurrence after CA, as shown in a recent meta-analysis [170]. They are also associated with reduced hospitalizations and ischemic stroke risk. A population study confirmed lower AF recurrence post-ablation in type 2 diabetes patients on SGLT2i [171].

Mechanistically, SGLT2i affects atrial electrophysiology by modulating sodium (INa), potassium (Ito, IKs), and calcium (ICa-L) currents, leading to reduced arrhythmogenicity [172].

### 9.8. Glucagon-like-Peptide-1 Receptor Agonists

Emerging data support the use of glucagon-like-peptide-1 receptor agonists (GLP-1 RAs) in reducing AF recurrence post-CA, particularly in patients with type 2 diabetes and obesity [173]. A recent meta-analysis confirmed this association [174]. Animal models show that GLP-1 RAs prevent atrial fibrosis, electrical remodeling, and conduction abnormalities [175].

### 9.9. n-3 Polyunsaturated Fatty Acids (PUFAs)

PUFAs have shown potential in reducing POAF, with greater benefit seen when the EPA/DHA ratio is <1 [176]. However, other studies have reported increased AF recurrence risk, especially at higher doses [177].

PUFAs offer multiple protective effects, including anti-inflammatory, anti-thrombotic, and antihypertensive properties, as well as favorable myocardial remodeling and electrophysiology [178]. In canine models, omega-3 PUFAs reduced atrial fibrosis, apoptosis, and stress-related proteins in the left atrium [179]. Further research is needed to clarify their role in AF prevention post-CA.

## 10. Future Therapeutic Targets

Targeting the complex mechanisms underlying AF—including inflammation, oxidative stress, and RAAS activation—may lead to novel therapies. Potential approaches include modulating inflammatory mediators such as HSP27, HSP60, IL-10, and the NLRP3 inflammasome, or using antibodies against DAMPs [180]. However, translating these targets into effective therapies remains a challenge.

Genetic polymorphisms affecting inflammatory responses suggest that personalized therapies based on individual inflammatory profiles may be necessary. Atrial-specific potassium channels like IKur and IK, ACh have been studied in phase II/III trials but have not demonstrated sufficient efficacy [168].

The TASK1 potassium channel is a promising target. TASK1 inhibition via small molecules or gene therapy has shown antiarrhythmic effects in preclinical studies [181]. Canakinumab, an anti-IL-1β monoclonal antibody, did not significantly reduce AF recurrence after ECV but did reduce AF burden [182].

Several additional agents show promise in preclinical models, though discussion of these is beyond the scope of this review and is detailed elsewhere [183].

## 11. Limitations of Current Research and Future Directions

Despite considerable advances, existing studies exhibit notable limitations, including small sample sizes, heterogeneity in patient populations, and insufficient long-term follow-up data. The molecular pathways leading to structural and electrical remodeling remain incompletely understood, particularly the precise roles and interactions of TGF-β1, matrix metalloproteinases (MMPs), and connexins in atrial fibrosis and conduction abnormalities. Moreover, inflammatory mediators, oxidative stress, and autonomic dysregulation in AF recurrence have yet to be fully characterized in clinical populations.

Future research must prioritize comprehensive mechanistic studies integrating genomics, proteomics, and advanced imaging to clarify these molecular pathways. Larger, multicenter trials with diverse patient cohorts and standardized outcome measures are essential to validate biomarkers such as serum TGF-β1, MMPs, connexins, and novel circulating markers like GDF-15 and relaxin. Additionally, exploring emerging risk factors like microplastic-induced inflammation could unveil new intervention targets. Ultimately, advancing the molecular understanding of AF recurrence will enhance patient selection, improve ablation strategies, and foster the development of personalized therapeutic approaches.

## 12. Conclusions

In summary, significant molecular mechanisms contributing to AF recurrence following catheter ablation include structural remodeling and fibrosis mediated by TGF-β1 and MMPs, ion-channel dysregulation, inflammatory responses, autonomic nervous system imbalance, and genetic and epigenetic alterations. Each of these mechanisms holds considerable clinical relevance, as they not only influence the likelihood of AF recurrence but also provide actionable targets for enhancing therapeutic efficacy. As clinical practice evolves, integrating molecular insights through precision diagnostics and tailored therapeutic interventions holds immense promise. Future clinical approaches should incorporate comprehensive biomarker profiling and personalized treatment strategies to refine patient selection, optimize ablation procedures, and improve long-term outcomes, ultimately reducing AF recurrence and associated cardiovascular complications (Table 1).

## Figures and Tables

**Figure 1 cells-15-00036-f001:**
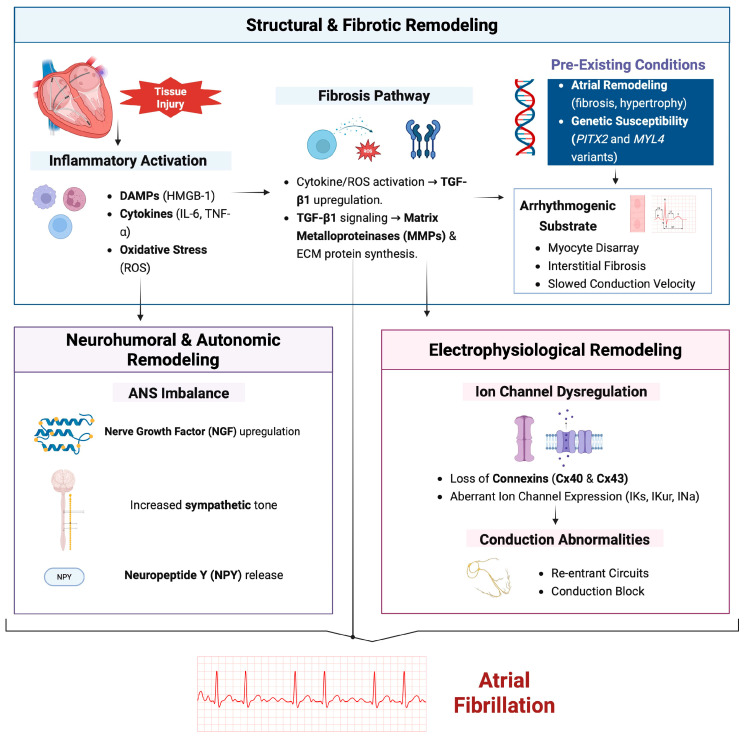
Mechanisms of atrial fibrillation: structural, neurohumoral, and electrophysiological remodeling. Created in BioRender. Mylavarapu, M. (2025) https://BioRender.com/5nc7afi (accessed on 20 December 2025).

**Table 1 cells-15-00036-t001:** Mechanism key findings.

Mechanism	Key Findings
Fibrosis and ECM Alterations	Atrial fibrosis significantly contributes to AF recurrence, involving irreversible structural remodeling compared to reversible electrical remodeling.
	TGF-β1 promotes atrial fibrosis by inducing collagen synthesis and ECM alterations through classical SMAD signaling pathways.
	Variations in TGF-β1 receptor interactions differentiate atrial from ventricular fibrosis.
Role of TGF-β and MMPs	Elevated plasma and serum TGF-β1 concentrations positively correlate with atrial fibrosis severity and AF recurrence.
	Matrix metalloproteinases (MMPs) and their inhibitors regulate ECM remodeling; elevated MMP-2 levels persist post-ablation and predict recurrence.
	Aggressive ablation strategies targeting structurally remodeled atria (e.g., CFAE ablation) have higher recurrence rates.
Cellular Hypertrophy & Myocyte Disarray	Cellular hypertrophy and disarray, especially in hypertrophic cardiomyopathy (HCM), increase recurrence risk by causing thicker atrial walls, diffuse fibrosis, and triggering ectopic activity beyond pulmonary veins.
	Combining catheter ablation with anti-arrhythmic drugs improves outcomes in HCM patients, despite high recurrence risk.
Conduction Velocity & Arrhythmogenic Substrate Formation	Functional conduction abnormalities (e.g., slow conduction velocity in left atrium) are predictive of AF recurrence post-ablation.
	Slow conduction velocities indicate structural remodeling, fibrosis, and atrial strain, predisposing to re-entry circuits and persistent AF.
Ion-Channel Dysregulation	Ion-channel dysfunction contributes to conduction slowing and re-entry circuit formation, pivotal to maintaining persistent AF.
Connexins and Gap Junctions	Connexin (Cx40, Cx43) remodeling disrupts electrical coupling, facilitating arrhythmogenic substrates and increased recurrence.
	Therapeutic modulation (e.g., beta-blockers) targeting gap junction integrity reduces recurrence risk.
Inflammatory Pathways	Post-ablation inflammation involving cytokines (CRP, TNF-α, IL-6) and DAMPs promotes fibrosis, structural remodeling, and AF recurrence.
Autonomic Nervous System Imbalance	ANS imbalances enhance susceptibility to AF recurrence through autonomic remodeling, affecting atrial electrophysiology.
Genetic & Epigenetic Factors	Specific genetic variants and epigenetic changes (DNA methylation, histone modifications) influence AF susceptibility and recurrence risk post-ablation.
Neurohumoral Factors & Biomarkers	Elevated circulating biomarkers (GDF-15, relaxin, NT-proBNP, and hs-CRP) correlate strongly with structural remodeling and AF recurrence.
Nanoplastics & Microplastics (MNPs)	MNPs induce systemic inflammation, oxidative stress, and endothelial dysfunction, all critical mechanisms in cardiovascular disease progression.
	Although direct evidence linking MNP exposure and AF recurrence post-ablation is lacking, the established role of inflammation and oxidative stress in AF supports further research into potential connections.

## Data Availability

Not applicable.

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
