# Peer review of "Molecular Mechanisms of Atrial Fibrillation Recurrence After Successful Catheter Ablation"

_cells, 2025, doi:10.3390/cells15010036_

Round 1
Reviewer 1 Report
Comments and Suggestions for Authors
This review tackles an important and timely topic—molecular determinants of AF recurrence after apparently successful ablation—and assembles a broad catalogue of candidate pathways (fibrosis/TGF-β–MMP axis, conduction heterogeneity, connexins, inflammation/oxidative stress, ANS remodeling, genetics/epigenetics, and circulating biomarkers), along with a therapeutic panorama (colchicine, RAS blockade/SV, SGLT2i, GLP-1 RAs, etc.). The following concerns should be resolved:
- For TGF-β1/MMPs, clarify whether signals predict recurrence independently of LA size/scar/LVAs and comorbidity burden, and whether they track PV reconnection vs extra-PV substrate. Consider a causal diagram that positions fibrosis mediators upstream of conduction slowing and LVAs, and shows plausible confounding/mediation.
- You nicely discuss slow LACV, pivot/slow zones, and activation patterns (“U-pattern” vs “D-pattern”). Please link these metrics to fibrosis imaging (LGE-MRI), voltage mapping, and outcomes after substrate-guided strategies vs PVI-only. Explicitly state generalizability across mapping systems and pacing protocols.
- For CRP/IL-6/GDF-15/relaxin, specify sampling time points (pre-ablation, 24–72 h, blanking period, ≥3 months), the temporal trajectory after RF vs cryo vs PFA, and whether early rises predict late recurrence after the blanking period. A figure with biomarker time-courses mapped to recurrence windows would help.
- Please clarify how ANS markers (NPY, SP) relate to clinical endpoints, and whether GP ablation or neuromodulation modifies these biomarkers contemporaneously with outcome. State if data are animal vs human and acknowledge gaps.
- Given rapidly expanding PFA use, discuss whether molecular injury and inflammatory cascades differ across RF, cryo, and PFA and how that might shift recurrence mechanisms/biomarker kinetics—and whether any data show different modulators (e.g., colchicine signal in RF vs PFA).
- The microplastics/nanoplastics paragraph feels disconnected from the ablation-recurrence narrative; either corroborate with cardio-electrophysiology-specific data or move to a brief “emerging exposures” box with explicit caveats.
Author Response
Response to Reviewer 1
Comment 1. For TGF-β1/MMPs, clarify whether signals predict recurrence independently of LA size/scar/LVAs and comorbidity burden, and whether they track PV reconnection vs extra-PV substrate. Consider a causal diagram that positions fibrosis mediators upstream of conduction slowing and LVAs, and shows plausible confounding/mediation.
Response:
We agree that this needed clearer causal framing. In the section on TGF-β1 and matrix metalloproteinases, we have:
- We clarified independence from LA size and comorbidities. We now explicitly state that several studies showed serum TGF-β1 or MMP-9 remained independent predictors of AF recurrence after adjusting for left atrial diameter and clinical risk factors in multivariable models.
- We have expanded the discussion to explain PV reconnection vs. extra-PV substrate.
Comment 2. You nicely discuss slow LACV, pivot/slow zones, and activation patterns (“U-pattern” vs “D-pattern”). Please link these metrics to fibrosis imaging (LGE-MRI), voltage mapping, and outcomes after substrate-guided strategies vs PVI-only. Explicitly state generalizability across mapping systems and pacing protocols.
Response:
We have expanded the section on conduction metrics to:
- Explain that regions with slow conduction and pivot/slow zones often overlap with LGE-MRI fibrosis and left-atrial low-voltage areas.
- Clarify that these observations appear consistent across high-density mapping systems.
- Discussed generalizability across mapping systems and pacing protocols.
Comment 3. For CRP/IL-6/GDF-15/relaxin, specify sampling time points (pre-ablation, 24–72 h, blanking period, ≥3 months), the temporal trajectory after RF vs cryo vs PFA, and whether early rises predict late recurrence after the blanking period. A figure with biomarker time-courses mapped to recurrence windows would help.
Response:
We have substantially clarified the timing and interpretation of inflammatory and neurohumoral biomarker measurements:
- We now summarize sampling time points, blanking period, and the temporal trajectory after RF vs cryo vs PFA.
- We clarify that GDF-15 and relaxin were mostly studied at baseline or in the early post-procedural phase, and that higher baseline levels have been associated with worse outcomes.
- We discussed biomarker kinetics differences of RF vs cryoballoon ablation vs PFA.
Comment 4. Please clarify how ANS markers (NPY, SP) relate to clinical endpoints, and whether GP ablation or neuromodulation modifies these biomarkers contemporaneously with outcome. State if data are animal vs human and acknowledge gaps.
Response:
In the Section Neurohumoral Factors and Circulating Biomarkers, we have:
- Expanded the text to explain that higher circulating NPY levels have been associated with AF presence, progression, and post-operative AF in human studies, and that increased NPY is linked to larger LA size and symptom burden.
- Discussed experimental work in canine models suggests long-term changes in NPY signaling after circumferential PV ablation may influence AF inducibility.
- Clarified that small human studies of neuromodulation have shown changes in NPY-related markers and cardiac glial protein S100B that correlate with rhythm outcomes.
- Explicitly acknowledge that, at present, NPY and SP are promising mechanistic biomarkers of autonomic remodeling, but not yet validated tools for predicting AF recurrence after catheter ablation.
Comment 5. Given rapidly expanding PFA use, discuss whether molecular injury and inflammatory cascades differ across RF, cryo, and PFA, and how that might shift recurrence mechanisms/biomarker kinetics—and whether any data show different modulators (e.g., colchicine signal in RF vs PFA).
Response:
We added a new subsection, “Ablation energy sources and their biological footprint”, which:
- Compares ablation techniques and discusses differences in molecular injuries and inflammatory cascades.
Because the evidence remains limited, we present these points as plausible hypotheses and emerging trends, not firm conclusions, and we explicitly label them as such in the text.
Comment 6. The microplastics/nanoplastics paragraph feels disconnected from the ablation-recurrence narrative; either corroborate with cardio-electrophysiology-specific data or move to a brief “emerging exposures” box with explicit caveats.
Response:
We agreed that the original paragraph on microplastics and nanoplastics felt somewhat abrupt. We have therefore:
- Moved and expanded this material into a new short subsection under “Emerging Exposures” rather than leaving it embedded in the main mechanistic narrative with caveats.
Reviewer 2 Report
Comments and Suggestions for Authors
Dear Authors,
The Authors provide a comprehensive description of the molecular mechanisms of atrial fibrillation recurrence after successful catheter ablation. They highlight structural remodeling and fibrosis modulated by TGF-beta and matrix metalloproteinases. This study also highlights other factors that increase the risk of AF, including cellular hypertrophy and myocyte disarray, ion channel dysregulation, electrophysiological and structural factors, and others. The authors do not limit themselves to structural remodeling but also highlight the role of the inflammatory pathway, autonomic nervous system imbalance, epigenetic and genetic factors, as well as neurohumoral factors and circulating biomarkers. This concise review also includes indications, therapeutic implications, and, crucially, future therapeutic targets. In summary, I highly recommend this study. I would like to point out some minor shortcomings:
- It seems reasonable to supplement the manuscript with other miRs (not only miR 1) that demonstrate profibrotic and antifibrotic effects (see paper: J. Clin.
Med. 2021, 10, 4430) - In Fig. 1, the font size is too small to read the figure accurately, and
- The literature requires refinement, e.g., references 124 and 125 are in red, and references 133 to 155 have the citation number written twice.
After corrections, the work is suitable for publication.
Author Response
Response to Reviewer 2
Comment 1. It seems reasonable to supplement the manuscript with other miRs (not only miR 1) that demonstrate profibrotic and antifibrotic effects (see paper: J. Clin.
Med. 2021, 10, 4430)
Response:
We amended the microRNA section to:
- We incorporated the suggested review by Sygitowicz et al. (J Clin Med 2021;10:4430), concluding that it is directly relevant and adds mechanistic depth to our discussion of microRNA.
Comment 2. In Fig. 1, the font size is too small to read the figure accurately
Response:
We have revised Figure 1 by enlarging the font size. The overall structure of the figure is unchanged; only the visual clarity has been improved.
Comment 3. The literature requires refinement, e.g., references 124 and 125 are in red, and references 133 to 155 have the citation number written twice.
Response:
We carefully reviewed the reference list and:
- All references have been re-checked for relevance and formatting. We eliminated duplicate citation numbers; the reference list has been renumbered accordingly.
Round 2
Reviewer 1 Report
Comments and Suggestions for Authors
The authors have appropriately revised their manuscript